# Physiological Benefits of Viewing Nature: A Systematic Review of Indoor Experiments

**DOI:** 10.3390/ijerph16234739

**Published:** 2019-11-27

**Authors:** Hyunju Jo, Chorong Song, Yoshifumi Miyazaki

**Affiliations:** 1Center for Environment, Health, and Field Sciences, Chiba University, Kashiwa, Chiba 277-0882, Japan; hyunju.jo@chiba-u.jp (H.J.); crsong@kongju.ac.kr (C.S.); 2Department of Forest Resources, Kongju National University, Yesan-gun, Chungcheongnam-do 32439, Korea

**Keywords:** nature therapy, visual effect of nature, natural environments, physiological relaxation, stress recovery, cerebral activity, autonomic nervous activity, heart rate variability, blood pressure, preventive medicine

## Abstract

Contact with nature has been proposed as a solution to achieve physiological relaxation and stress recovery, and a number of scientific verification outcomes have been shown. Compared with studies of the other senses, studies investigating the visual effects of nature have been at the forefront of this research field. A variety of physiological indicators adopted for use in indoor experiments have shown the benefits of viewing nature. In this systematic review, we examined current peer-reviewed articles regarding the physiological effects of visual stimulation from elements or representations of nature in an indoor setting. The articles were analyzed for their stimulation method, physiological measures applied, groups of participants, and outcomes. Thirty-seven articles presenting evidence of the physiological effects of viewing nature were selected. The majority of the studies that used display stimuli, such as photos, 3D images, virtual reality, and videos of natural landscapes, confirmed that viewing natural scenery led to more relaxed body responses than viewing the control. Studies that used real nature stimuli reported that visual contact with flowers, green plants, and wooden materials had positive effects on cerebral and autonomic nervous activities compared with the control. Accumulation of scientific evidence of the physiological relaxation associated with viewing elements of nature would be useful for preventive medicine, specifically nature therapy.

## 1. Introduction

For approximately 6–7 million years, human beings have evolved in the natural environment [1]. Therefore, the physiological functions of human bodies developed in response to the natural environment [2]. The start of the industrial revolution in the UK in the 18th century led to the rapid growth of industrialization and urbanization and significantly changed our living environments. Urban environments, which account for a mere 0.01% of dwelling environments in human history, remain an unfamiliar environment for the human body. In 2008, more than half of the world’s population lived in urban areas and by 2050, 69% of humans will live in urban areas [3,4].

Since the late 20th century, the restorative effect of nature has been gradually gaining attention in the fields of environmental psychology and public health [5,6,7,8,9,10,11,12,13,14,15]. Environmental psychologists have discussed aesthetic and affective responses to the outdoor environment and the preference for natural scenery over urban landscapes, which lack natural elements [6,7,8]. In 1984, Ulrich reported that surgical patients who were assigned to rooms with windows overlooking natural scenery had shorter hospital stays and experienced fewer negative health outcomes than patients in rooms with windows facing the brick wall of a building [9].

In recent years, nature therapy has been proposed as one of the solutions for stress recovery and health promotion [16]. Nature therapy is defined as “a set of practices aimed at achieving ‘preventive medical effects’ through exposure to natural stimuli that render a state of physiological relaxation and boost the weakened immune functions to prevent diseases” [17]. With the increasing social attention given to health and the development of physiological measurement devices, researchers have been collecting scientific data on nature therapy. Several study outcomes have shed light on the benefits of nature therapy, such as physiological relaxation and immunity function improvement, by direct exposure to natural environment [18,19,20,21,22,23,24,25,26,27,28,29,30,31,32] and natural elements (flowers [33,34,35,36,37,38], green plants [39,40,41,42,43,44,45,46,47,48], or wooden materials [49,50,51,52,53,54,55,56,57,58]) or by indirect exposure to natural environments through a display [59,60,61,62,63,64,65,66,67] or projector [68,69,70,71,72,73]).

In particular, indoor experiments, where it is possible to control the stimuli and physical environment, entail more specific and diverse methodological approaches than those used in field experiments. Of the five senses, the effectiveness of sight [33,34,35,39,40,41,42,43,44,45,46,47,48,54,55,56,57,58,59,60,61,62,63,64,65,66,67,68,69,70,71,72,73,74], smell [36,37,38,52,53,75] and touch [49,50,51] have been discussed. The rapid development of physiological indicators has enabled evaluation of body responses such as cerebral activity (functional magnetic resonance imaging [fMRI], near-infrared spectroscopy [NIRS] and electroencephalography [EEG]), autonomic nervous activity (heart rate variability [HRV], heart rate, pulse rate and blood pressure) and endocrine activity (salivary cortisol concentration). Such data could be helpful in understanding the mechanisms underlying the physiological responses emerging from contact with nature; furthermore, the data would be useful for understanding the different effects shown by field [19,20,21,22,23,24,25,26,27,28,29,30,31,32] and indoor experiments [33,34,35,36,37,38,39,40,41,42,43,44,45,46,47,48,49,50,51,52,53,54,55,56,57,58,59,60,61,62,63,64,65,66,67,68,69,70,71,72,73,74,75]. In particular, compared with studies that examine the effects of the other senses, studies on the visual effects of nature are at the forefront of research. Researchers have accumulated data in this field since 1981 [6].

Therefore, in this systematic review, we summarized peer-reviewed articles that discussed the physiological effects of visual stimulation from real elements or representations of nature determined by indoor experiments. The purpose of this study was to understand the entirety of this research domain and to discuss the potential of nature therapy as a type of preventive medicine.

## 2. Materials and Methods

This systematic review was conducted in accordance with PRISMA (the guidelines of Preferred Reporting Items for Systematic reviews and Meta-Analyses) statement guidelines [76]. Eligible studies required: (1) a sample of healthy people or patients, (2) intervention of natural elements or scenery, (3) comparison of non-natural elements or scenery, or another type of natural scenery [62,71,72], and (4) outcome of physiological response. We reviewed the scientific literature on the visual physiological effects of nature by searching the PubMed database, which provides scholarly articles related to the medical and health fields. The publication date was set from 1 January 1970 to 31 July 2019, and the date of the last search was 27 August 2019. We used a combination of terms to search the literature related to (1) nature, (2) physiological effects and (3) visual effects. The keywords related to (1) nature were “landscape”, “forest”, “urban park”, “green”, “plant”, “wood”, “foliage”, and “flower”; the keywords related to (2) physiological effects were “physiological”, “brain”, “prefrontal”, “autonomic nervous”, “stress”, “immune system”, “endocrine”, “cortisol”, “immunoglobulin”, “blood pressure”, “heart rate” and “pulse rate”; and the keywords related to (3) visual effects were “visual” and “view”. To retrieve specifically relevant literature from the database, we set a limit to the search range within the titles and abstracts, and we searched articles about the human species and for journals written in English. The inclusion criteria for studies in the review were (1) studies conducted in an indoor setting (2) studies that presented visual stimuli and (3) peer-reviewed journal papers. The exclusion criteria were (1) review articles, (2) abstract-only articles, and (3) physiological indicator-lacking articles.

One reviewer (H.J.) screened the title, abstract, and full text of each article to identify those that were eligible for the systematic review, according to the inclusion and exclusion criteria mentioned above. Thereafter, the selected full-text articles were screened by another reviewer (Y.M.). Disagreements between reviewers were resolved by discussion to achieve consensus.

## 3. Results

Figure 1 shows the systematic review flow chart. We found 733 relevant articles in the literature that included the abovementioned keywords in the titles and abstracts. At first screening based on title and abstracts, 29 articles were obtained. After a full-text reading, 15 articles were considered eligible articles. We also reviewed and added 22 articles relevant to the present review topic found in the references of the articles. In total, we found 37 articles that presented evidence on the physiological effects of viewing nature. The articles collected in this study were reviewed in detail, particularly the measures, visual stimulation type, experimental protocol and results. We classified the studies on the basis of the stimulation method used to examine physiological effects, nature display stimuli and real nature stimuli. Between 1981 and 2019, a variety of findings were published on the physiological effects of viewing nature stimuli (summarized in Table 1). In some studies, the participants were exposed to images of natural elements, such as slides, photos, or videos of forests and/or urban park landscapes. In others, they were exposed to real (life) nature stimuli, such as green plants, flowers and/or wooden materials.

### 3.1. Physiological Effects of Viewing Images of Nature Stimuli

Laboratory studies comparing the effects of viewing photos and/or videos of natural environments or elements with those of viewing controls have demonstrated the physiological recovery effects of natural settings. Most studies have reported improvements in stress and fatigue recovery and relaxation effects from viewing natural settings by using a range of objective physiological indicators.

#### 3.1.1. Physiological Effects of Viewing a Forest Landscape

Research has validated information regarding some ideas that we all intuitively know; for example, that we feel relaxed and calm in a forest environment. A number of studies have reported that our bodies have positive physiological responses to forest environments. Although science supports these claims, there has been growing interest and attention in health promotion for forest therapy, or shinrin-yoku [16,18]. The studies below indicated that people felt relaxed in forest environments [59,62,73] and felt benefits immediately when exposed to forest interior environments with surrounding trees and vegetation [71].

Song et al. reported that visual stimulation from viewing a forest scene produced activity in the brain and autonomic nervous system. Seventeen female university students viewed a forest scene and an urban landscape scene (control) on a large plasma display window for 90 s while their oxy-Hb concentrations in the prefrontal cortex and HRV were continuously measured [59]. The results showed that visual stimulation by viewing the forest scene induced decreased oxy-Hb concentrations in the right prefrontal cortex.

Another study investigated the physiological relaxation effect related to the individual preference of nature scenes [62]. Twelve male adults each viewed their individual preferred video of a sea or forest, and their physiological changes, as shown by HRV and heart rate, were recorded. The participants were divided into two groups of six on the basis of their preference for sea or forest scenery, and each physiological change indicator was compared among the groups. The heart rate while viewing the sea video was higher than that while viewing forest video. In the same year, one study examined physiological relaxation effects to determine if forest locations and vegetation density affect human attention and relaxation state [71]. One hundred and eighty university students were recruited in the study, and the participants were placed in groups of six (*n* = 30) that participated in a visual stimulation experiment. Pictures of three locations representing a forest interior (featuring surrounding trees and vegetation), a forest edge (featuring the visibility of the overlapping patches) and a forest exterior (featuring where the patch can be seen from far away) as well as three pictures of vegetation with different densities (high, medium and low) were collected. Six types of visual stimulation slides were eventually prepared. The participants watched the slideshow, and their electroencephalographic alpha frequency, which is associated with a state and alert relaxation, was measured. The results showed that the forest interior group had higher alpha values than those of the forest edge group, which suggested that the interior group was less relaxed than the edge group.

Duncan et al. compared the effects of green exercise and the effects of exercise alone (control) on blood pressure and heart rate [73]. Fourteen primary school children were asked to cycle for 15 min while watching videos of either forest settings or a blank screen. Consequently, compared with the participants in the control group, those in the green exercise group showed significant decrease in systolic blood pressure.

#### 3.1.2. Physiological Effects of Viewing an Urban Green Landscape

Because of advantages such as easy access from their dwelling space, many urban residents visit urban green spaces, such as parks or gardens, for recreation, restoration and healing. In accumulating scientific evidence on the positive effects of natural environments, some researchers have recently reported on the health benefits of these urban green spaces, for example, viewing urban green scenes, such as garden and urban parks, decreased brain activity and stress [61,63,64]. Lee has recently reported on the visual effects of garden landscapes compared with those of city landscapes [61]. In this indoor experiment, 18 healthy young adults participated (nine males, nine females), and the oxy-Hb concentration in the prefrontal cortex, systolic and diastolic blood pressure and pulse rate were used as indicators of autonomic nervous activity measured while the participants viewed garden and city landscape images. Winter landscape images of Anapji, one of the traditional gardens in Korea, were chosen as a garden landscape visual stimulation, and images of a commercial area of the city center were chosen as a control stimulation. A total of 12 landscape images of each stimulation, taken from three different distances (near, medium and far), were presented for 90 s each. The oxy-Hb concentrations of the left and right prefrontal cortex were significantly lower while viewing garden landscape images than those while viewing city landscape images. A significant difference was found between males and females as follows: female participants showed a decrease in the oxy-Hb concentrations while viewing both landscapes, whereas male participants showed a distinct difference when viewing city landscapes (an increase) and garden landscapes (a decrease). Among the autonomic nervous changes, diastolic blood pressure showed different responses from males (a decrease for garden landscapes) and females (an increase for garden landscapes). The findings of the study verified that gender may be a factor in producing different physiological effects as well as relaxation effects when viewing urban green spaces. Another study compared the stress-recovery effects of different scenes of urban parks with those of scenes of urban roads [63]. After completing an oral exam as a stressor, 70 male and 70 female university students watched videotaped scenes during a stress-recovery stage while having their stress and attention levels measured by skin conductance response (SCR) and heart rate. The study found that the perceived restorative effects, such as decreased SCR and heart rate, were more significantly revealed for two scenes depicting a lawn without people and a small lake compared with a paved-plaza scene with or without people. The study confirmed that an urban green landscape scene without people was more restorative than was a scene depicting people and that nature-based components were more likely to reduce stress than were hardscape components. In 2015, visual effects of urban green spaces were verified using the indicators of parasympathetic and sympathetic nervous activities [64]. After undergoing a psychological stress task, 46 university students were exposed to urban park scenes and urban-built scene, which was shown through a computer screen, for 5 min. As a result, restorative effects were observed after viewing urban green scenes, as marked by a significant increase in respiratory sinus arrhythmia (RSA), indicating parasympathetic nervous activity.

#### 3.1.3. Physiological Effects of Viewing Mixed Natural Landscapes

The physiological benefits of viewing natural scenery have been of great interest both theoretically and empirically. In particular, many experimental studies in the field of environmental psychology have shown the restorative potential of natural environments by using videos, photographs and natural landscape stimulation of mountains, fields, water and forests. The studies showed that viewing images of natural landscapes, whether presented as 2D photographs, or virtual reality (VR) scenes, resulted in restorative effects, such as deactivating visual and attentional areas of the brain, reducing eye blinking and relieving stress [6,8,10,60,65,66,67,68,69,70,72,74].

The technique of fMRI provides an opportunity to further explore the psychophysiological benefits of viewing natural environments. This novel approach enables direct examination of regional brain activity while viewing landscapes. Tang et al. compared the restorative value of four types of landscape environments (urban, mountain, forest and water) by using fMRI to investigate regional brain activity [60]. The study recruited 39 adults between the age of 20 and 30 years, and data from 31 participants (14 males and 17 females) were used after excluding for movement artifacts. The nature landscape included images from three types of common natural setting: mountains, forests and water. Urban landscapes were retrieved from an online gallery of photographs. In total, 12 photographs, three from each of the four categories, with similar color, lightness and layout, were used in the experiment. Compared with the mountain and water landscapes, urban images increased visual and attentional focus, which resulted in activation of the cuneus. This finding indicates that viewing mountain and water landscapes after urban landscapes may reduce activation in the visual cortex and increase activation in the rest of the attention system. Compared with urban landscapes, water landscapes were associated with increased neural activation in the attention area of the brain, which suggested that viewing water landscapes may stimulate the rest of the attention system. Interestingly, a lack of significant difference in brain activities between viewing urban and forest landscapes suggests a smaller effect on attention restoration than viewing the mountain and water landscapes. Overall, different landscapes affected regional brain activity differently; most notably, the visual and attention areas of the brain responded differently to images of urban and natural environments.

In another study, the brain activity of 28 right-handed participants triggered by visual stimuli, such as natural scenic views (i.e., natural landscapes, mountains, parks, and forests) and urban scenic views (i.e., city landscapes and tall buildings), was recorded by fMRI [65]. Brain areas that were predominantly activated by natural scenic views included the superior and middle frontal gyri, superior parietal gyrus, precuneus, basal ganglia, superior occipital gyrus, anterior cingulate gyrus, superior temporal gyrus, and insula. Conversely, brain areas that were activated in response to urban scenic views primarily comprised the middle and inferior occipital gyrus, parahippocampal gyrus, hippocampus, amygdala, anterior temporal pole, and inferior frontal gyrus. These findings suggest that the differential functional neuroanatomies for each scenic view type can be elucidated by emotional responses to natural and urban environments.

Similarly, in another study during the same year, fMRI was used to evaluate the brain activity of 30 college students with rural (13.5 ± 5.7 years) and urban (13.8 ± 4.4 years) life experiences [66]. Rural scenes included forests, gardens, parks, and hills, whereas urban scenes included apartments, buildings, electrical cables, and factories. Different brain areas were activated while viewing the rural and urban scenes. The superior parietal gyrus, anterior cingulate gyrus, postcentral gyrus, globus pallidus, putamen, and caudate nucleus head were mainly involved while viewing the rural scenes. In contrast, the hippocampus, parahippocampal gyrus, amygdala, and lingual gyrus were primarily involved while viewing the urban scenes. These findings demonstrated an improved characterization of neural activation, indicating that a nature-oriented lifestyle is inherently preferred.

In 2008, some researchers verified the restorative effects of viewing a natural scenery on the basis of “Attention Restoration Theory” using the indicators of EEG and blood volume pulse (BVP) [68]. In total, 110 adults participated in this study. Compared with viewing a scene of a solid blue screen, viewing a nature scene, such as wildland, increased the alpha wave and decreased the BVP; an increase of BVP indicates a decrease in sympathetic nervous activity. Thus, a natural environment such as wildland had an impact on physiological restoration.

Researchers often use the autonomic nervous system, which is vital in homeostasis and normal and stress-responsive physiology, to understand the physiological effects of nature. In 2003, a research reported heart rate response to natural and urban environments [67]. In this study, 28 female students watched a nature video depicting a waterside and an urban video depicting pedestrian streets. Consequently, the heart rate of the participants significantly decreased after viewing the nature video. In 2012, Gladwell et al. also explored autonomic nervous system control during the viewing of nature and built scenes. They measured heart rate, HRV and blood pressure [69]. Slides of nature and urban scenes were projected onto a screen (1.8 m × 1.8 m) while the participants lay in a semisupine position. Standard deviation of RR intervals (SDRR), reflecting overall HRV, and parasympathetic nervous activity increased significantly during the viewing of nature images relative to those during viewing built environment scenes, which suggested increases in vagal activity. This finding suggests that the simple act of viewing nature may induce changes in autonomic control.

The following year, Brown et al. examined the effects of viewing nature scenes on the recovery of autonomic activity after the introduction of a stressor [70]. Heart rate, HRV and systolic and diastolic blood pressure were used as markers of autonomic function change, and physiological data from 23 adults were collected. The study involved showing the participants 20 photographs of nature and built scenes in the same order for 10 min (each photograph for 30 s) in a Microsoft PowerPoint slideshow. Before seeing the photographs, the participants performed a forward digit span test with an accompanying social-evaluative threat to elicit cardiovascular stress. SDRR was significantly greater in viewing nature scenes than built scenes. The finding shows that viewing images of nature improved the recovery process following a stressful event.

Some researchers have applied three-dimensional (3D) images or VR technology to the field of psychology and healthcare. Using VR is an efficient way to introduce natural settings to participants who could not otherwise access nature. A research compared the effects of 3D and two-dimensional (2D) images of nature. The prefrontal cortex and sympathetic nervous activities of 19 male university students were continuously evaluated by NIRS and HRV while they looked at 3D and 2D images of water lily [72]. The right prefrontal cortex activity and sympathetic nervous activity significantly decreased while the participants looked at the 3D rather than the 2D images of water lily; thus, compared with the 2D image, the realistic 3D image of nature induced physiological relaxation to a greater extent.

In 2017, Anderson et al. reported on the relaxation effects of nature scenes presented by using VR [74]. Eighteen healthy adults viewed 360° natural scenes for 15 min after performing a stressful metal task. The nature scenes included a large expansive natural view of water, sheep, birds, houses and beaches. The participants’ electrodermal activity (EDA) and HRV were recorded consistently, and physiological changes between viewing nature and control (empty classrooms) scenes were compared. The EDA decreased more after viewing natural scenes than control scenes, which indicated that viewing natural scenes through VR can lead to relaxation effects.

With the use of videos of nature-dominated and artifact-dominated drives, the stress-recovery effects of a roadside environment were assessed in a study [8]. After engaging in a stressful task, 160 university students watched drive-view videos for 10 min. Compared with exposure to artifact-dominated views, exposure to nature-dominated views, such as forests and fields, significantly decreased the EDA activity and systolic and diastolic blood pressures.

Using EEG, Ulrich measured the brain activity and heart rate of 18 adults while watching water-containing nature scenes, vegetation-dominated nature scenes, and urban environment images. Alpha waves were higher when the participants viewed the water-containing nature scenes rather than when viewing the urban environment images [6]. Moreover, 10 years later, Ulrich et al. assessed the physiological responses of participants to natural and urban settings and specifically considered that the SCR, heart rate, and pulse transit time were associated with visual stimuli. Furthermore, natural visual stimuli decreased the heart rate and SCR, demonstrating that exposure to a natural environment has a restorative influence [10].

### 3.2. Physiological Effects of Viewing Real Nature Stimuli

#### 3.2.1. Physiological Effects of Viewing Green Plants

Many studies have reported the effect of viewing green plants on physiological relaxation. The findings confirmed that viewing green plants, such as foliage plants, in an indoor environment can elicit positive health outcomes with greater stabilization of prefrontal cortex activity and autonomic nervous activity [39,40,41,42,43,44,45,46,47,48].

Oh et al. recently reported on the visual impact of real-foliage plants (i.e., non-patterned *Epipremnum aureum*) in comparison with artificial plants, a plant photograph, and no plant, on 23 elementary students [39]. These students were evaluated using an EEG during exposure to each visual stimulus for 3 min. Results showed that viewing real plants significantly decreased the theta waves of the frontal lobe. Therefore, viewing living plants can improve the concentration of elementary students.

Another study considered the physiological effects of viewing a scene of a bamboo plant, which is a well-known and popular choice in Chinese landscape design, on 40 Chinese university students (20 males and 20 females) [41]. The participants’ blood pressure was measured using an EEG while they observed a potted bamboo plant or an empty pot (control) for 3 min. Significantly lower systolic and diastolic blood pressures and higher alpha and beta waves were observed in the participants who viewed the bamboo plant compared with those who viewed the empty pot. Hence, the visual stimulation provided by the bamboo plant induced physiological relaxation.

In 2016, Park et al. investigated the effects of viewing foliage plants on prefrontal cortex activity. In a crossover test, 24 male university students stared at a container with and without foliage plants (*Epipremnum aureum*) for 3 min [42]. While the participants viewed the container, the oxy-Hb concentration in their prefrontal cortexes was measured by continuously using a portable NIRS. Compared with the no-foliage condition, after viewing foliage plants for 3 min, the oxy-Hb concentration in the right prefrontal areas of the participants decreased significantly within the first minute. Brain blood flow is consistent with the level of oxy-Hb, so a decrease in oxy-Hb concentration equates to a physiological relaxation effect. In 2017, the same team of researchers investigated prefrontal cortex activity and HRV in 24 male university students [44]. The participants were asked to transfer 500 g pots with or without a foliage plant (*Peperomia obtusifolia*) to a tray. Although this task involved tactile stimulation, the only difference in visual sensation was the presence or absence of a plant. Each subject transferred approximately 33 pots over a 3-min period at a self-controlled speed. The ln [low frequency (LF)/high frequency (HF)] value, which represents sympathetic nervous activity, was significantly lower in the task performed with foliage plants compared with that without foliage plants. This finding suggested that the visual stimulation with foliage plants resulted in physiological relaxation.

Ikei et al. used changes in the pulse rate and HRV of 85 high school students (41 males, 44 females) as indexes of autonomic nervous activity to examine the physiological effects of viewing foliage plants [46]. Visual stimuli comprised three foliage plants with linear leaf patterns (*Dracaena deremensis*). The participants viewed a scene with foliage plants or a same scene without foliage plants for 3 min. Clear between-group differences were observed. The mean pulse rate significantly decreased by 0.1% during visual exposure to foliage plants relative to the pulse rate under the control condition. In addition, during visual exposure to foliage plants, there was a significant increase in the HF component of HRV, a marker of parasympathetic nervous activity, and a significant decrease in the LF/(LF + HF) ratio, a marker of sympathetic nervous activity. These findings indicated that the autonomic nervous activity was alleviated by visual exposure to foliage plants. These findings provide evidence supporting the physiological benefits of viewing green plants.

Regarding the physiological benefits of viewing real-foliage green plants, a study found that direct visual contact with real-foliage plants was more important and more effective than visual contact made through images of foliage plants [45]. The researchers verified the physiological effects assessed by oxy-Hb concentrations in the prefrontal cortex in 18 female university students while the participants viewed a real plant or a projected image of the same plants. Viewing actual foliage plants significantly increased oxy-Hb concentrations in the left and right prefrontal cortex, whereas viewing the projected images did not.

Bonsai have been well-known as indoor plants that stimulate self-induced mental relaxation among Japanese people. Two recent studies on nature therapy confirmed that viewing bonsai induced physiological relaxation in Japanese people [40,43].

One study was performed on 14 elderly patients undergoing rehabilitation [40]. While each participant was watching a bonsai or control condition (nothing) for 1min, his/her NIRS, HRV, and pulse rate were measured. The results showed that visual stimulation of bonsai increased in parasympathetic nervous activity and decreased in sympathetic nervous activity. These two researches verified the physiological relaxation effects of viewing bonsai among Japanese patients who have high mental stress and physical pains.

In the other study relevant to physiological relaxation of a bonsai, 24 Japanese male patients with spinal cord injuries who had experienced mental and physical stress viewed a miniature potted 10-year-old Japanese cypress bonsai (40 × 20 × 5 cm), and the changes in oxy-Hb concentration in the prefrontal cortex (NIRS) and HRV were used as indexes of physiological relaxation [43]. When the patients viewed the bonsai, the oxy-Hb concentration in their left prefrontal cortex significantly decreased, and the HF components of HRV (which is related to parasympathetic nervous activity) increased, whereas the LF/HF ratio of HRV (which is related to sympathetic nervous activity) increased relative to that of the patients who viewed an empty desk as a control.

There have been clinical trials that explored the health benefits of viewing indoor plants on the recovery of surgical patients within hospital rooms. In 2008, Park and Mattson performed a randomized clinical trial with hemorrhoidectomy surgery patients on the effects of foliage plants in hospital rooms [47]. Researchers evaluated physiological relaxation through medical measurements such as length of hospitalization, use of analgesics for postoperative pain control and vital signs (systolic and diastolic blood pressures, heart rate and respiratory rates). The patients were randomly assigned to a control room or a plant room that had identical interiors except for the absence or presence of plants. All patients were in good health before their diagnosis and surgical treatment. In the plant room, 12 potted foliage and flowering plants were placed with sterile soilless potting mix. The patients in the plant group were allowed to view plants during their recovery periods after surgery until they were discharged. Compared with the control group, the plant group was less frequently given weak and moderate analgesics. On both the day of surgery and first recovery day, the systolic blood pressure and heart rate were significantly lower in the plant group than in the control group. The following year, Park and Mattson again verified the recovery effects of viewing indoor plants through clinical trials with hemorrhoidectomy surgical patients [48]. The procedures and measurements of the experiment were identical to their 2008 study. This time, only systolic blood pressure showed a significant difference, which was still lower in the plant group than in the control group on the first day after surgery. The findings of the two studies confirmed that introducing plants into a hospital room during the recovery period had a positive influence linked directly to physiological benefits for surgical patients.

#### 3.2.2. Physiological Effects of Viewing Flowers

The physiological effects of viewing fresh roses, which belong to one of the most popular flower species, have been reviewed. The results verified the idea that viewing fresh roses stimulates physiological relaxation and that fresh flower are more effective than artificial flowers [33,34,35].

One recent study by Song et al. investigated the heart rates, HRV and physiological changes in the activity of the right and left prefrontal cortex activity of 15 female university students while they were exposed to visual stimulation with roses [33]. In the experiment, 25 fresh, unscented, red, roses trimmed to a length of 40 cm were arranged in a glass vase on a desk, and each participant was asked to sit and stare at the roses for 3 min. Their physiological responses were assessed and compared with those elicited under the control conditions, which involved staring at an empty desk for 3 min. The results showed that viewing fresh roses reduced oxy-Hb concentration in the right prefrontal cortex in the first minutes, which verified the idea that visual stimulation with roses relieved prefrontal activity. Additionally, there was a marginally significant decrease in ln[LF / (LF + HF)], a marker of sympathetic nervous activity, during visual stimulation with roses.

The other study was conducted with office workers because it has been suggested that they have easy access to nature, such as flower arrangements, despite the time and space constraints under which they work [34]. Researchers measured the pulse rates and change in HRV of 31 male office workers. The researchers compared the difference in physiological parameters between viewing a scene with fresh pink roses and viewing the same scene without roses. During the 4-min exposure time, the high-frequency component mean value, indicating parasympathetic nervous activity, was significantly higher during visual stimulation by roses than under the control condition. This finding indicated that viewing fresh roses induced physiological relaxation in the office workers. Two other studies verified that viewing fresh roses led to physiological relaxation.

Another study verified the physiological effects of viewing fresh flowers by comparing their effect to that of artificial flowers [35]. The researchers compared the physiological effect of viewing fresh yellow pansies with that of viewing artificial yellow pansies made of a polyester material in 40 high school students. The seated participants were exposed to visual stimulation for 3 min, and their pulse rate and HRV were measured by using fingertip accelerated plethysmography. HRV showed that viewing of real pansies reduced the sympathetic nerve activity more than viewing artificial pansies.

#### 3.2.3. Physiological Effects of Viewing Wooden Materials

Some research has used natural stimuli, including forests, urban parks, flowers and foliage plants, to investigate the relaxation effect induced by viewing nature. Some researchers have focused on the effects of wooden materials, which are the natural stimuli to which we are most often exposed in daily life. These physiological findings demonstrated the visual relaxation effect of wooden environments. The physiological effects were also associated with differences in the quantity of wood and individual preferences [54,55,56,57,58]. Most studies have adopted visual stimuli on a large scale, such as actual-size wooden interior rooms or wooden panel-covered walls. The studies reported that the wooden rooms induced physiological relaxation effects not observed in non-wooden rooms, and some reported that the percentage of wood or the design of a wooden room induced different physiological changes.

Recent research by Zhang et al. investigated physiological responses to wooden and non-wooden indoor environments [54]. The researchers prepared three wooden rooms with different coloring as follows: (1) 100% dark brown wood, (2) 50% light brown wood and 50% painted white, (3) 100% light brown wood and (4) a non-wooden room (painted white) as a control. The study adopted the autonomic nervous system, respiratory system, and visual system as the physiological indicators. Twenty healthy adults participated in the experiment and were asked to complete work tasks in one of the wooden rooms or the non-wooden room; their exposure time was about 90 min each. Consequently, their systolic blood pressure and heart rate were lower in the wooden rooms than in the non-wooden room, and the oxyhemoglobin saturation (SpO_2_) levels were higher in the wooden rooms than in the non-wooden room. These physiological findings demonstrated the visual relaxation effect of the wooden environments. In 2005, Sakuragawa et al. investigated the autonomic nervous responses of wood by using full-sized hinoki wall panels and a white steel wall panel as a control [55]. In this study, the systolic blood pressure decreased in the subjects who preferred the hinoki wall panels, whereas it increased in the subjects who disliked the white steel wall panel.

In 2002, Tsunetsugu et al. verified the visual effects of wooden materials through three experiments conducted in actual-size model rooms with different designs and wood ratios [56]. They used the blood pressure and pulse rate of 10 participants as physiological indicators. Each participant sat in a wheelchair, entered each experimental room and observed its interior for 90 s. The results showed that the pulse rates were lower in the standard room that had wood flooring and papered walls and ceiling relative to the pulse rates in the room designed with wooden beams and columns. Three years later, Tsunetsugu et al. investigated the changes in the brain and autonomic nervous system responses in the same experimental setting [57]. During this investigation, both the standard room and the designed rooms resulted in significant increase in the left prefrontal cortex activity; furthermore, pulse rates decreased in the standard room but increased in the designed room. Lastly, in 2007, the researchers examined the physiological effects of actual-size model rooms with wood ratios of 0%, 45%, and 90% [58]. The prefrontal cortex activity increased in the 0% and 90% wood ratio rooms. The systolic blood pressures decreased for the first part of the stimulation in the room with a 90% wood ratio, and the diastolic blood pressures decreased regardless of wood ratio. Conversely, pulse rate increased in the 45% and 90% wood ratio rooms.

## 4. Discussion

The present study aimed to understand the entirety of this research domain and to review the potential of nature therapy as a type of preventive medicine. The 37 articles on the physiological effects of viewing nature were reviewed in detail. The study using nature images (or videos) as stimuli investigated the physiological effects of viewing a large range of natural landscapes, such as forests, urban greens and general natural landscapes with multiple natural elements (e.g., mountain, trees, fields, and water). The studies on real nature stimuli investigated the effects of viewing specific natural elements (e.g., flowers, indoor green plants and wooden materials).

A review on the health effects of viewing landscape in environmental psychology reported that the natural landscapes gave a stronger positive health effect compared to urban landscapes [77]. In this review, most studies related to displaying images of natural landscapes compared the physiological effects of city/built landscape images and confirmed that relaxed body responses occurred while viewing natural landscape scenery as well. For example, urban images activated the visual and attentional areas of the brain that nature landscape did not [65], oxy-Hb concentrations in the prefrontal area decreased while viewing forest scenes [59] and autonomic nervous responses, such as parasympathetic activity, were significantly higher during the recovery steps after viewing nature scenes [64]. The studies that used real nature stimuli reported that visual contact with green plants and flowers in daily environments had positive impacts, such as decreased sympathetic nerve activity and increased parasympathetic nerve activity [33,34,35,40,43,44,46].

The use of VR techniques and 3D images to maximize visual effects could offer more realistic interesting visual stimulations than those of 2D photographs. Some studies that investigated this topic found that EDA decreased more after viewing natural scenes by using VR techniques than by viewing control scenes, which indicated that viewing natural scenes through VR could lead to relaxation effects [74]. Igarashi et al. reported that a realistic 3D image of nature induced a significant decrease in the prefrontal cortex activity, whereas a similar 2D image did not [72].

Although the majority of the studies consistently showed positive effects from exposure to nature scenery and settings surrounded by nature, some studies obtained different results. For example, contact with real nature aroused prefrontal areas and increased the pulse rate [57,58]. The findings of physiological responses could be different than those of other studies because of improperly designed experimental protocols, or a large difference between individuals.

On the other hand, some researchers mentioned individual differences in physiological responses to the relaxation effects of nature. In review articles on nature therapy, Song et al. and Ikei et al. described individual differences in physiological relaxation effects and proposed that physiological responses could vary depending on the participants’ personalities [17,78,79,80]. Accordingly, future verification of the physiological differences according to age, sex and individual personality would be needed to further elucidate the physiological pathways leading from viewing nature to relaxation.

Although most studies in this review included healthy adults from the general public, more research on different groups, in particular on highly stressed groups such as depressed patients, elderly rehabilitation patients, or hypertensive individuals who may show higher stress levels in our modern society, would be useful. Contact with nature may be a preventive and restorative public health strategy, particularly for individuals at higher risk of ill health [81].

Bodily reactions during contact with nature have been described by investigations using a variety of physiological indicators (e.g., NIRS, EEG, fMRI, HRV, heart rate and blood pressure). These findings have strengthened and deepened the growing evidence base on the health benefits of nature. To improve the health of modern people and encourage greater use of nature therapy in preventive medicine, a better understanding of the physiological effects of viewing nature through continued research is important.

### Limitations

This present review has some limitations. First, the review protocol was not registered in an international database such as PROSPERO. A future systematic review should be registered in such a database. Second, the evaluated articles were obtained from only a single database, but an important requirement for future studies is the use of multiple databases for a comprehensive search across interdisciplinary topics. Third, this study did not perform a risk of bias assessment or a study quality assessment. It is essential that risk of bias evaluations (e.g., bias in the stimulation orders) are implemented in experimental studies. Future systematic reviews should, therefore, involve such assessments of the risk of bias and study quality. Fourth, some of the studies compared the best sides of nature with the worst sides of non-natural environments. To generalize the effects of nature, it is a necessary to compare general natural and non-natural environments. Finally, most of the presented articles employed small samples, which could have affected scientific validity. In future, researchers should adopt larger and more specific samples that include individuals of different ages and genders.

## 5. Conclusions

The study examined 37 articles presenting evidence of the physiological effects of viewing nature based on indoor setting experiments—the studies were classified based on the stimulation method used. Exposures to display stimuli, such as photos, 3D images, virtual reality, and videos of nature, confirmed that viewing natural scenery induced a more physiological relaxing state than viewing the control. Furthermore, exposure to real stimuli, such as green plants, flowers, and wooden materials, had positive effects on the prefrontal cortex and autonomic nervous activities compared with the control. These findings have strengthened and deepened the growing evidence base of the health benefits of nature. Accumulation of that scientific evidence supports the idea that nature therapy would be useful for preventive medicine.

## Figures and Tables

**Figure 1 ijerph-16-04739-f001:**
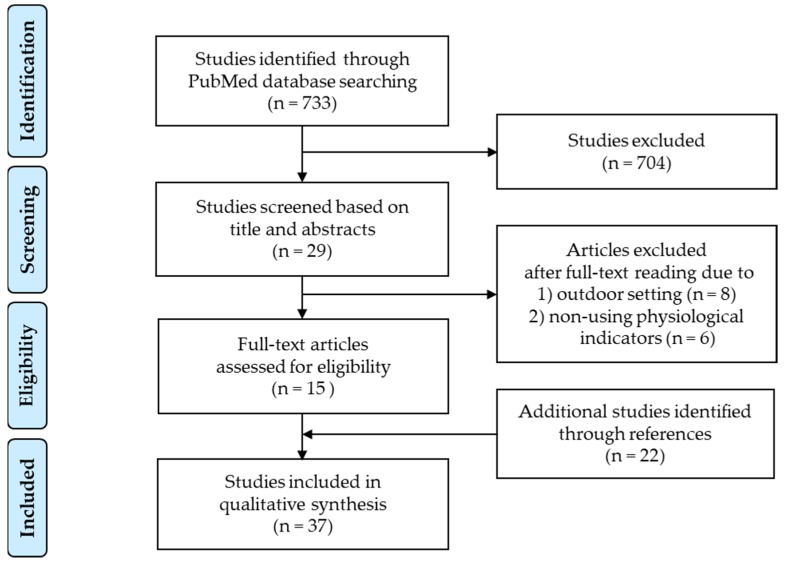
Systematic review flow chart [76].

**Table 1 ijerph-16-04739-t001:** Overview of Indoor Experiments on Visual Physiological Effects of Nature.

Year	Author	Stimulation/Control	Stimulation Time	Stimulation Method	Summary	Participants	Article Type	Ref No.
2019	Oh et al.	Actual foliage plants/No plants, artificial plants, photograph of plants	3 min	Real (Foliage plant)	(1) Brain activity 1) EEG: Decrease in theta wave * comparison with control	Elementary students *N* = 23	Original article	[39]
2018	Song et al.	Bonsai/Without bonsai	60 s	Real (Bonsai)	(1) Brain activity (prefrontal cortex activity) 1) NIRS: No significant change (2) Autonomic nervous activity 1) HRV • Parasympathetic nervous activity: Increase • Sympathetic nervous activity: Decrease 2) Purse rate: No significant change * comparison with control	Elderly patients (undergoing rehabilitation) *N* = 14 (4 males, 10 females)	Original article	[40]
2018	Song et al.	Forest/City	90 s	Display (Forest)	(1) Brain activity (prefrontal cortex activity) 1) TRS: Decrease in the right prefrontal activity (2) Autonomic nervous activity 1) HRV: No significant change * comparison with control	Female univ. students *N* = 17	Original article	[59]
2017	Hassan et al.	Pot with bamboo plants/Pot without bamboo plants	3 min	Real (Bamboo plant)	(1) Brain activity 1) EEG: Increase in alpha and beta waves (2) Autonomic nervous activity 1) Blood pressure • Systolic blood pressure: Decrease • Diastolic blood pressure: Decrease 2) Purse rate: No significant change * comparison with control	Univ. students *N* = 40 (20 males, 20 females)	Letter to the editor	[41]
2017	Park et al.	Pot with foliage plant/Pot without foliage plant * task: transferring pots to a tray	3 min	Real (Foliage plant)	(1) Brain activity (prefrontal cortex activity) 1) NIRS: No significant change (2) Autonomic nervous activity 1) HRV • Parasympathetic nervous activity: No significant change • Sympathetic nervous activity: Decrease * comparison with control	Male univ. students *N* = 24	Original article	[44]
2017	Song et al.	Rose flowers/Without rose flowers	3 min	Real (Rose flower)	(1) Brain activity (prefrontal cortex activity) 1) TRS: Decrease in the right prefrontal activity (2) Autonomic nervous activity 1) HRV • Parasympathetic nervous activity: No significant change • Sympathetic nervous activity: Marginally decrease (*p* < 0.06) 2) Heart rate: No significant change * comparison with control	Female univ. students *N* = 15	Original article	[33]
2017	Ochiai et al.	Bonsai/Without bonsai	60 s	Real (Bonsai)	(1) Brain activity (prefrontal cortex activity) 1) NIRS: Decrease in the left prefrontal activity (2) Autonomic nervous activity 1) HRV • Parasympathetic nervous activity: Increase • Sympathetic nervous activity: Decrease * comparison with control	Adult male patients with spinal cord injury *N* = 24	Original article	[43]
2017	Lee	Garden landscape/City landscape	90 s	Display (Garden landscape)	(1) Brain activity (prefrontal cortex activity) 1) NIRS: Decrease in the left and right prefrontal activity (2) Autonomic nervous activity 1) Blood pressure: No significant change 2) Pulse rate: No significant change * comparison with control	Adults *N* = 18 (9 males, 9 females)	Original article	[61]
2017	Anderson et al.	VR natural scenes of mixed nature (Ireland) and beach (Australia)/Empty indoor classroom * after stress stimulation of arithmetic test	15 min	VR (Beach)	(1) Autonomic nervous activity 1) EDA: Decrease in viewing natural scenes (beach) 2) HRV: No significant change * comparison with control	Adults *N* = 18 (9 males, 9 females)	Original article	[74]
2017	Chiang et al.	Forest pictures of locations (interior, edge, exterior) and vegetation density (high, medium, low) * using 3D glasses * after mental stress task	6 min 30 s	3D projector (Forest)	(1) Brain activity 1) EEG • Increase in alpha wave in the forest interior * comparison with forest edge	Univ. students *N* = 180 (82 males, 98 females)	Original article	[71]
2017	Zhang et al.	Wooden rooms/Non-wooden room	60 min	Real (Wooden room)	(1) Autonomic nervous activity 1) Blood pressure: Decrease in systolic blood pressure 2) Heart rate: Decrease 3) SpO2: Increase 4) Skin temperature: No significant change * comparison with control	Univ. students *N* = 20 (10 males, 10 females)	Original article	[54]
2017	Tang et al.	Mountain, water, forest/Urban built space	30 s	Display (Mountain, forest, water)	(1) Brain activity 1) fMRI • Mountain: Contrast is revealed in the left and right cuneus • Water: Contrast is revealed in the left and right cuneus; Activations of the right cingulate gyrus and left precuneus • Forest: No activation * comparison with control	Adults *N* = 31 (14 males, 17 females)	Original article	[60]
2017	Tsutsumi et al.	Sea video, forest video * exposure to both visual and sound	15 min	Display (Forest, sea)	(1) Autonomic nervous activity 1) Heart rate: Forest < Sea	Adult males *N* = 12	Original article	[62]
2016	Park et al.	Pot with foliage plants/Pot without foliage plants	3 min	Real (Foliage plant)	(1) Brain activity (prefrontal cortex activity) 1) NIRS: Decrease in the right prefrontal activity * comparison with control	Male univ. students *N* = 24	Original article	[42]
2016	Wang et al.	Urban park/Roadway * videos	8 min	Display (Urban park)	(1) Autonomic nervous activity 1) SCR: Decrease 2) Heart rate: Decrease * comparison with control	Univ. students *N* = 140 (70 males, 70 females)	Original article	[63]
2015	Van den Berg et al.	Urban park/Urban built space * after psychological stress test	5 min	Display (Urban park)	(1) Autonomic nervous activity 1) RSA (indicator of parasympathetic nervous activity): Increase 2) PEP (indicator of sympathetic nervous activity): No significant change * comparison with control	Univ. students *N* = 46 (21 males, 25 females)	Original article	[64]
2015	Igarashi et al.	Fresh/Artificial yellow pansies	3 min	Real (Pansy flower)	(1) Autonomic nervous activity 1) HRV • Parasympathetic nervous activity: No significant change • Sympathetic nervous activity: Decrease 2) Pulse rate: No significant change * comparison with control	High school students *N* = 40 (19 males, 21 females)	Original article	[35]
2014	Igarashi et al.	Real-foliage plants/Display of foliage plants	3 min	Real (Foliage plant)	(1) Brain activity (prefrontal cortex activity) 1) NIRS: Increases in the left and right prefrontal activity * comparison with control	Female univ. students *N* = 18	Short communi-cation	[45]
2014	Ikei et al.	Foliage plants/Without foliage plants	3 min	Real (Foliage plant)	(1) Autonomic nervous activity 1) HRV • Parasympathetic nervous activity: Increase • Sympathetic nervous activity: Decrease 2) Pulse rate: Decrease * comparison with control	High school students *N* = 85 (41 males, 44 females)	Original article	[46]
2014	Igarashi et al.	3D image/2D image of water lily	90 s	3D projector (Water lily)	(1) Brain activity (prefrontal cortex activity) 1) NIRS: Decrease in the right prefrontal activity (2) Autonomic nervous activity 1) HRV • Parasympathetic nervous activity: No significant change • Sympathetic nervous activity: Decrease * comparison with control	Male univ. students *N* = 19	Short report	[72]
2014	Duncan et al.	Forest/Blank screen * cycling watching video	15 min	Projector (Forest)	(1) Autonomic nervous activity 1) Blood pressure • Systolic blood pressure: Decrease • Diastolic blood pressure: No significant change 2) Heart rate: No significant change * comparison with control	Children *N* = 14 (7 males, 7 females)	Commu- nication	[73]
2014	Ikei et al.	Rose flowers/Without rose flowers	4 min	Real (Rose flower)	(1) Autonomic nervous activity 1) HRV • Parasympathetic nervous activity: Increase • Sympathetic nervous activity: No significant change 2) Pulse rate: No significant change * comparison with control	Male office workers *N* = 31	Original article	[34]
2013	Brown et al.	Tree, grass etc./Urban built space * after mental stress task	10 min	Projector (Tree, grass etc.)	(1) Autonomic nervous activity 1) HRV • SDRR (reflects overall HRV): Increase • Parasympathetic nervous activity: No significant change 2) Herat rate: No significant change 3) Blood pressure: No significant change * comparison with control	Adults *N* = 23 (6 males, 17 females)	Original article	[70]
2012	Gladwell et al.	Tree, grass etc./Urban built space	5 min	Projector (Tree, grass etc.)	(1) Autonomic nervous activity: 1) HRV • SDRR (reflects overall HRV): Increase • Parasympathetic nervous activity: Increase 2) Blood pressure: No significant change 3) Heart rate: No significant change * comparison with control	Univ. students & staffs *N* = 29	Original article	[69]
2010	Kim et al.	Forest, park, garden etc./Urban built space	2 min	Display (Forest, park, garden etc.)	(1) Brain activity 1) fMRI • Natural scenic views: Activations of the superior and middle frontal gyri, superior parietal gyrus, precuneus, basal ganglia, superior occipital gyrus, anterior cingulate gyrus, superior temporal gyrus, and insula • Urban scenic views: Activations of the middle and inferior occipital gyri, parahippocampal gyrus, hippocampus, amygdala, anterior temporal pole, and inferior frontal gyrus * comparison with pre-stimulation	Adults *N* = 28 (16 males, 12 males)	Original article	[65]
2010	Kim et al.	Forest, park, garden etc./Urban built space	2 min	Display (Forest, park, garden etc.)	(1) Brain activity 1) fMRI • Nature scenic photos: Activation of the anterior cingulate gyrus, globus pallidus, putamen, and head of the caudate nucleus • Urban scenic photos: Activation of the hippocampus, parahippocampus and amygdala * comparison with pre-stimulation	Univ. students *N* = 30 (18 males, 12 males)	Original article	[66]
2009	Park and Mattson	Hospital rooms with/Without foliage and flowering plants	3 days	Real (Foliage, flowering plant)	(1) Autonomic nervous activity 1) Blood pressure • Systolic blood pressure: Decrease • Diastolic blood pressure: No significant changes 2) Heart rate: No significant changes 3) Body temperature: No significant change 4) Respiratory rates: No significant change * comparison with control	Patients (recovering from a hemorrhoidec-tomy) *N* = 90	Original article	[48]
2008	Park and Mattson	Hospital rooms with/Without foliage and flowering plants	3 days	Real (Foliage, flowering plant)	(1) Autonomic nervous activity 1) Blood pressure • Systolic blood pressure: Decrease • Diastolic blood pressure: No significant change 2) Heart rate: Decrease 3) Body temperature: No significant change 4) Respiratory rates: No significant change * comparison with control	Patients (recovering from an appendec-tomy) *N* = 90	Research report	[47]
2008	Chang et al.	Wildland scene/Non-viewing slide consisting of solid blue scene	10 s	Projector (Wildland)	(1) Brain activity 1) EEG: Increase in alpha wave (2) Autonomic nervous activity 1) BVP (an increase indicates decrease in sympathetic nervous activity): Decrease * comparison with control	Adults *N* = 110	Original article	[68]
2007	Tsunetsu-gu et al.	Actual-size model rooms (wood ratios: 0%, 45%, 90%)	90 s	Real (Wooden room)	(1) Brain activity (prefrontal cortex activity) 1) NIRS: Increase in right prefrontal activity in the 0%, 90% wood ratio rooms (2) Autonomic nervous activity 1) Blood pressure • Systolic blood pressure: Decrease in the 90% wood ratio room • Diastolic blood pressure: Decrease in the 0%, 45%, and 90% wood-ratio rooms 2) Pulse rate: Increase in the 45% and 90% wood ratio * comparison with pre-stimulation	Male univ. students *N* = 15	Original article	[58]
2005	Tsunetsu-gu et al.	Standard room with wooden flooring and wall and ceiling covered by white wallpaper/Designed room with wooden beams and columns	90 s	Real (Wooden room)	(1) Brain activity (prefrontal cortex activity) 1) NIRS: Increase in the left prefrontal activity in the designed and the standard rooms (2) Autonomic nervous activity 1) Blood pressure • Systolic blood pressure: No significant change • Diastolic blood pressure: No significant change 2) Pulse rate • Decrease in the standard room • Increase in the designed room * comparison with pre-stimulation	Male univ. students *N* = 15	Original article	[57]
2005	Sakuraga-wa et al.	Full-sized Japanese cypress wall panels/White steel wall panels	90 s	Real (Wooden wall)	(1) Autonomic nervous activity 1) Blood pressure • Systolic blood pressure: Decrease in subjects who liked hinoki wall panel • Diastolic blood pressure: No significant change 2) Pulse rate: No significant change * comparison with pre-stimulation	Male univ. students *N* = 14	Original article	[55]
2003	Laumann et al.	River, sea/Urban built space * videos	20 min	Display (River, sea video)	(1) Autonomic nervous activity 1) Heart rate: Decrease * comparison with control	Female univ. students *N* = 28	Original article	[67]
2002	Tsunetsu-gu et al.	Standard room with wooden flooring and wall and ceiling covered by white wallpaper/Designed room with wooden beams and columns	90 s	Real (Wooden room)	(1) Autonomic nervous activity 1) Blood pressure: No significant change 2) Pulse rate: Standard room < Designed room	Male univ. students *N* = 10	Original article	[56]
1998	Parsons et al.	Forest, field/Urban built space * drives view videos * after stress task	10 min	Display (Forest, field)	(1) Autonomic nervous activity 1) EDA: Decrease 2) Blood pressure • Systolic blood pressure: Decrease • Diastolic blood pressure: Decrease * comparison with control	Univ. students *N* = 160 (81 males, 79 females)	Original article	[8]
1991	Ulrich et al.	Tree, stream/Urban built space * videos	10 min	Display (Tree, stream)	(1) Autonomic nervous activity 1) Heart rate: Decrease 2) SCR: Decrease 3) Pulse transit time (shorter times are associated with higher systolic blood pressure): Increase * comparison with control	Univ. students *N* = 120 (60 males, 60 females)	-	[10]
1981	Ulrich	Water, vegetation/Urban built space	13 min	Projector (Water, vegetation)	(1) Brain activity 1) EEG: Increase in alpha waves (2) Autonomic nervous activity 1) Heart rate: No significant change * comparison with control	Adults *N* = 18 (9 males, 9 females)	-	[6]

Note: EDA, Electrodermal activity; EEG, Electroencephalography; fMRI, functional magnetic resonance imaging; HRV, Heart rate variability; NIRS, Near-infrared spectroscopy; TRS, Near-infrared time-resolved spectroscopy; RSA, Respiratory sinus arrhythmia; SCR, Skin conductance response; SDRR, Standard deviation of RR intervals; SpO_2_, Oxyhemoglobin saturation; PEP, Pre-ejection period; VR, Virtual reality.

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
