# Peer review of "Physiological Benefits of Viewing Nature: A Systematic Review of Indoor Experiments"

_ijerph, 2019, doi:10.3390/ijerph16234739_

Round 1
Reviewer 1 Report
This paper deals with important research themes. If this paper is published, it will contribute to many future studies and practices. However, there seem to be some deficiencies in this paper. That is a particular requirement in the PRISMA statement. I would like you to make corrections based on the comments below.
PRISMA reqaire a structured abstract. This is not necessarily the case because this journal does not require a structured abstract. However, methods should be described in a little more detail. (PRISMA # 2) Did you register your review protocol somewhere? Please add if you have (eg PROSPERO). If not, please state this as a limitation. (PRISMA # 5) If possible, please describe the Eligibility criteria in more detail, such as target age and region. (We recommend using PICO.) (PRISMA # 6) Please include the date of the last search by PubMed. (PRISMA # 7) Also, since no other electronic database is used at this time, please state in the Limitation that you searched using only a single database. It is important to use multiple databases for a comprehensive search. Especially for interdisciplinary themes like this. More documents than database searches are added by manual search. This can be seen as a result of insufficient search in the database. You have to Indicate information collected from accepted articles in the method (PRISMA # 11) If you have assessed bias risk, please describe the method and results. (PRISMA # 12, 19) If you have not done, you can do it (for example, use https://www.nhlbi.nih.gov/health-topics/study-quality-assessment-tools) . If you will not do it, please state in the limitation. Please consider including L87-92 and Fig 1 in the Result. (PRISMA # 17) How about changing "L95 3. Physiological Responses of Viewing Nature” to ”Results”? Please add a Limitaion subsection at the end of the Discussion. (PRISMA # 25) Ten. Please add Conclusion section (PRISMA # 26)Author Response
Please see the attachment.

Reviewer 2 Report
The submitted Review article has been developed with a consistent methodology and provides a good overview of the state of the arts for the studied topic: Physiological benefits of viewing nature in indoor experiments.
However and despite the comprehensive and synthetic presentation of different and relevant articles, the following suggestions are proposed for the consideration of the authors. These suggestions are based in the potential that review papers have to provide “a critical, constructive analysis of the literature in a specific field through summary, classification, analysis, comparison, etc” (Mayer 2009).
INTRODUCTION: Define more precisely the objective of the review article, the main challenges affecting the definition of a more coherent body of knowledge in the studied field and the Research Questions/challenges that will be answered/solved through the Review. Make more explicit the main goals of the review article (to evaluate literature, to identify patterns and trends in the literature, to synthesize literature, to identify research gaps and recommend new research areas, etc.) (Mayer, 2009) Concerning the previous point, it would be useful to determine if the existing literature can be organized more clearly and if this is one of the goals of the review article. For instance, according to the type of stimuli provided by nature (direct or real versus indirect or virtual, based in a single element or material versus based in a more complex experience), based in the use of different types of physiological indicators, etc.
MATERIALS and METHODS Methodologically, the article is based in a systematic review of recent literature. The method is clearly explained and the Figure 1 illustrates well the whole process. The selection of the PubMed database seems to be adequate for the identification of articles. However, the incorporation of some additional references from key authors in the field (Velarde, 2007; Maller, 2006, C. Ward-Thompson, 2001, etc.) could be beneficial for the development of the Introduction, Discussion and Conclusion.
REVIEW It would be important to indicate the criteria used to systematize the presentation of papers in the section 3. Now this information is presented in the Discussion (lines 447-450) but could be moved to the beginning of the Chapter 3. Furthermore, it would be important to explain why the authors decided to organize the review according to the stimulation method, which other alternatives they considered and why those other alternatives were discarded. Some subsections of the section 3 finish with a clear and concise conclusion (e.g. lines 285-287). These conclusions or findings can be extremely useful for the reader and perhaps should be included at the beginning of all the subsections as a sort of synthesis that is afterwards justified in the following paragraphs (“proving the finding” by summarizing and comparing the different articles included in each subsection).
DISCUSSION / CONCLUSION The Discussion section works also a “Conclusion”. Probably, the definition of a clearer objective and Research Questions/ Challenges in the introductory section would facilitate a more critical discussion and would help the reader to understand how the developed Review has systematized, synthetized or compared the existing literature and how it has detected the main gaps or the most prevalent lines of research in the studied field. One of the main contributions of the article is the classification proposed by the authors to organize the existing literature. It would be interesting to include a critical discussion about this contribution, for instance by considering how consistent that classification is in comparison to other possible classifications, its advantages and shortcomings, etc. The suggested lines of research are extremely interesting. However, in addition to the need of developing new research in more specific groups, it must be noted that most of the presented articles were dealing with very small samples. Does it affect their scientific validity?. It would be also interesting to discuss if the presented experiments compare the best sides of nature with the worst sides of non-natural environments. This can generate a general and qualitative bias. The presented articles and the final discussion seem to assume that people from different cultures, social groups, etc. have a similar response to nature. Accordingly, the beneficial effects of both real or virtual nature in human health and wellbeing would be based in a genetic cause rather than in cultural one. Do the authors share this opinion or would they suggest developing some research on the causes explaining the beneficial effects of being visually exposed to nature?
Reviewer 3 Report
It is an interesting review paper that suggests various consequences to public institutions. For this reson it is better to split out the discussion from the conclusion. It could underline better the results of this good research.
Round 2
Reviewer 1 Report
Thank you for your responses. This article has been corrected well. We would appreciate it if you could consider only the following points.
About PICO, what is shown in the table is the result. You should describe in the methods section in the text what type of patient, intervention, comparison, and outcome were targeted. Was the screening done independently by two or more people? If so, it is better to describe it to improve the quality of the article. If not, it is recommended that you describe how many people screened and the procedure. I couldn't point it out last time. I'm sorry.Author Response
Please see the attachment.
